# Analgesic and Sedative Effects of Epidural Lidocaine-Xylazine in Elective Bilateral Laparoscopic Ovariectomy in Standing Mule Mares

**DOI:** 10.3390/ani11082419

**Published:** 2021-08-17

**Authors:** Cecilia Vullo, Adolfo Maria Tambella, Marina Meligrana, Giuseppe Catone

**Affiliations:** 1Department of Chemical, Biological, Pharmaceutical, and Environmental Sciences (ChiBioFarAm), Via S. D’Alcrontes, 31 University of Messina, 98166 Messina, Italy; 2School of Bioscience and Veterinary Medicine, University of Camerino (MC), Via Circonvallazione 92-93, 62022 Matelica, Italy; adolfomaria.tambella@unicam.it; 3Veterinary Practitioner, Parghelia (VV), 89900 Vibo Valentia, Italy; marina.meligrana@hotmail.it; 4Department of Veterinary Sciences, University of Messina, Polo Universitario dell’Annunziata, 98168 Messina, Italy; gcatone@unime.it

**Keywords:** epidural anaesthesia, laparoscopy, mule mares, ovariectomy, pain control, sedation

## Abstract

**Simple Summary:**

Elective bilateral ovariectomy in mule mares is recommended to correct undesirable behaviour effects related to oestrus and also in order to improve work performance. This procedure is usually carried out in a standing position with a combination of sedative and analgesic drugs along with local anaesthesia of the ovarian pedicles. The aim of the study was to evaluate the effects of epidural lidocaine-xylazine in sedated standing mules undergoing elective bilateral laparoscopic ovariectomy. The findings suggest that no significant changes were found in heart rate, respiratory rate, rectal temperature with respect to baseline values. Sedation and analgesia were satisfactory in six out of the eight mules. Epidural analgesia with lidocaine-xylazine may be a good alternative to local anaesthetics infiltration of the ovarian pedicle for elective bilateral laparoscopic ovariectomy in standing mule mares, but further studies should be performed.

**Abstract:**

The purpose of this study was to determine the analgesic efficacy and safety of epidural lidocaine-xylazine administration in standing mules undergoing elective bilateral laparoscopic ovariectomy in order to suppress unwanted behaviour. Eight mule mares were sedated with intramuscular 0.05 mg/kg acepromazine followed by 1.3 mg/kg of xylazine and 0.02 mg/kg of butorphanol intravenously. Sedation was maintained by a constant rate infusion of 0.6 mg/kg/h of xylazine. The paralumbar fossae were infiltrated with 30 mL of 2% lidocaine. Epidural anaesthesia was performed at the first intercoccygeal space with 0.2 mg/kg of lidocaine and 0.17 mg/kg of xylazine. After 15 min, bilateral laparoscopic ovariectomy was performed. Heart rate, respiratory rate, rectal temperature, invasive arterial blood pressure, degree of analgesia, sedation and ataxia were evaluated during surgery. The laparoscopic ovariectomy was successfully completed in all animals. Sedation and analgesia were considered satisfactory in six out of the eight mules. In conclusion, caudal epidural block allowed surgery to be easily completed in six out of eight. The animals did not show any signs of discomfort associated with nociception and were mostly calm during the procedures, however additional studies are needed to establish epidural doses of xylazine and lidocaine that result in reliable abdominal pain control in mules for standing ovariectomy.

## 1. Introduction

Mule (Equus mulus), interspecific hybrids between a male donkey and a female horse, combine features from both donkey and horse, leading to great diversity in size, temperament and body type [1].

Although mule mares are sterile, they show a rather variable oestrus [2]. During this period, they may present typical unwanted modification of behaviour such as reluctance to work, aggression, kicking and biting [3].

Bilateral ovariectomy in mule mares is an elective procedure performed, as in equine mares, to suppress unwanted oestrus behaviour and to decrease visceral pain occurring during oestrus [4,5,6]. In addition, it is performed in order to improve work performance. The surgery may be carried out by laparotomy under general anaesthesia, or by laparoscopy which can be performed either in the standing sedated horse or in dorsal recumbency under general anaesthesia [7]. An advantage of standing sedation over general anaesthesia is the decreased risk of mortality [8].

The benefits of standing laparoscopy, compared to laparotomy performed under general anaesthesia, include lower costs, direct visualization of abdominal viscera, secure haemostasis, tension-free manipulation of the ovaries and smaller flank incisions. These advantages reduce complications and hospitalization time associated with ovariectomy [9,10]. To provide a good level of analgesia with no response to surgical stimulus, the use of a combination of sedative and analgesic drugs associated with locoregional technique is required [10]. In general, local analgesia is achieved by paravertebral anaesthesia and/or by the subcutaneous, intramuscular and subperitoneal administration of local anaesthetics bordering the dorso-caudal aspect of the last rib and the ventro-lateral aspect of the lumbar transverse processes. The analgesia is potentiated by local anaesthetic infiltration of the ovarian pedicles or by simple topical anaesthetic application to the ovary, which might reduce the risk of trauma caused by the needle which in turn can cause haemorrhage thereafter [11,12].

In horses, the epidural space is accessible at the lumbosacral joint (cranial epidural) or caudal to the sacrum (caudal epidural). The caudal epidural approach (sacrococcygeal or Co1–Co2 space) is preferable, as it is not only easier or safer to perform and reduces the potential for nerve damage with respect to subarachnoid anaesthesia. Indeed, the caudal approach reduce the risk of inadvertent injection in the subarachnoid space as well as the risk of motor blockade of the pelvic limbs, compared to the cranial approach. Caudal epidural anaesthesia is used to desensitize the coccygeal and last two or three sacral nerves, providing sensory loss of the anus, rectum, perineum, vulva, vagina, urethra, and bladder with the aim of inducing regional anaesthesia for surgery without losing motor function of the hind limbs [13]. This technique is also used to perform standing surgery using the flank approach to the abdomen [14,15]. The injection of drugs is performed into the extradural space, i.e., outside the dura mater, but underneath the ligamentum flavum [16]. Complications are represented by failure to achieve a good analgesia, ataxia or recumbency due to overdose and motor and sympathetic blockage [16,17]. A combination of a local anaesthetic drug with an alpha2-adrenergic agonist or an opioid is the most popular option, as this combination extends the duration of the epidural anaesthesia or analgesia in horses [18,19,20].

The purpose of this study was to determine the analgesic efficacy and safety of caudal epidural lidocaine-xylazine in standing mules undergoing elective bilateral laparoscopic ovariectomy.

## 2. Materials and Methods

The study was approved by the Bioethics Committee of the Department of Veterinary Sciences of the University of Messina and followed the Good Scientific Practice Guidelines and adhered to European legislation, EU Directive 2010/63 (Ethical approval code N. 050/2021).

Eight healthy adult mule mares, American Society of Anaesthesiologists (ASA) physical status 1, aged 3 to 7 years, with a body weight ranging between 380–450 kg, were included in the study. Exclusion criteria included ASA physical status ≥2, intractable behaviour, neurological or neuromuscular disease and skin infection at the site of the surgery or that of the epidural anaesthesia. Mules had free access to water, but food was withheld for 16 h prior to surgery. Sodium benzylpenicillin [30,000 U/kg, intramuscularly (IM)] in combination with gentamicin [6.6 mg/kg intravenously (IV)] and flunixin meglumine (1.1 mg/kg, IV) were administered preoperatively.

Body weight estimated on the bases of the body length and chest girth of the mules [21]. The mares were placed in a stock and the tails were wrapped (Benda PoweFlex, Alcyon, Italy).

No other physical restraint was provided. The animals were sedated by the same expert anaesthetist and the laparoscopic ovariectomy was performed on a farm where the animals were housed. Any eventual disturbing factors which could have contributed to a sudden recovery from sedation were avoided because the stock was located at an adequate distance from the boxes. Before administering any drug, baseline rectal temperature (RT, °C), heart rate (HR, beats/min), and respiratory rate (RR, breaths/min) were recorded. HR was measured by auscultation, RR by observing thoracic excursion and RT was measured using a digital thermometer. A 14-gauge, 13 cm catheter was aseptically placed in the external jugular vein, through a lidocaine bleb injected subcutaneously 5 min before. Mules were premedicated with 0.05 mg/kg acepromazine (Prequillan^®^, Fatro, Italy) IM and after 15 min 1.3 mg/kg of xylazine (Rompun^®^; Bayer, Italy) and 0.02 mg/kg of butorphanol (Dolorex^®^, Animal Health, Milan, Italy) were administered IV. Depth of sedation was evaluated before epidural administration using a 4-point sedation scale from Vullo et al., 2017 [22] (Table 1). Additional xylazine (0.3 mg/kg IV) was administered when the mules were poorly sedated (score 0). Sedation thereafter was maintained by a constant rate infusion (CRI) of 0.6 mg/kg/h of xylazine that was started when the degree of sedation was considered adequate (score 2 or 3).

The mules’ eyes were covered with a drape and cotton wool balls were placed within the external ear canals to minimize visual and auditory stimulation, respectively.

Prior to surgery, the left and right paralumbar fossae were clipped and aseptically prepared. The dorso-caudal aspect of the last rib and the ventro-lateral aspect of the lumbar transverse processes were infiltrated with 30 mL of 2% lidocaine (Lidocaina 2%^®^, Esteve, Milan, Italy). Following aseptical preparation, epidural anaesthesia was performed between the first and the second intercoccygeal vertebrae (Co1-Co2) space, which was identified by palpating the first moveable coccygeal articulation while raising and lowering the tail. A small skin bleb of lidocaine 2% was placed at the proposed injection site. A spinal needle (Terumo^®^, Rome, Italy) of 7.5 cm in length and 18 gauge thickness was introduced at a 30–45 degree angle into the specified space. The exact position of the needle in the space was verified by loss of resistance (LOR) before the injection. The epidural injection was performed administering 0.2 mg/kg of lidocaine (2%) and 0.17 mg/kg of xylazine (2%) diluted in 10 mL saline 0.9% (Sodium Chloride 09%, S.A.L.F.^®^, Bergamo, Italy). After drug administration, 0.5 mL saline solution was introduced into the hub to empty the residual anaesthetic. Afterwards, the tail was tied to the stock. Bilateral laparoscopic ovariectomy started 15 min after epidural administration. Once identified the anatomic landmarks for the three laparoscopic portals, another 5 mL lidocaine 2% was infiltrated at each side.

A 20- or 22-gauge catheter was aseptically placed in the facial or transverse facial artery and invasive systolic (SAP), diastolic (DAP) and mean arterial blood pressures (MAP) and HR, were recorded every 10 min during surgery using a multiparametric monitor (BeneView T8, Mindray, New York, NY, USA). RR and RT were always monitored clinically every 10 min. Depth of sedation and ataxia were scored 10 min after the epidural anaesthesia (T0). Analgesia, depth of sedation, and ataxia were scored from start of the surgery (T1), at the end of surgery (T6), using a scoring system modified from Sampaio et al., 2017 and Schauvliege et al., 2019 [23,24] (Table 2). Analgesia was evaluated starting from T1. If the quality of sedation (0 or 1) or analgesia (0 or 1) were considered inadequate, a supplemental IV bolus of xylazine 0.65 mg/kg was given. If the ataxia was considered inadequate, the constant rate infusion (CRI) of xylazine was stopped.

The drape was gently removed to assess the sedation score.

Using an electronic carbon dioxide insufflator, a pneumoperitoneum was created adding a pressure not exceeding 15 mmHg. A bilateral paralumbar fossa approach was used, using three portals per side, as described in previous study [25]. Both ovaries were then removed by extending the caudoventral incision on the left side dorsally to the initial primary incision.

If the mules responded to surgical stimuli during ovariectomy, the ovarian pedicle was infiltrated with 20 mL of lidocaine 2%.

Cardinal data were presented as mean ± standard deviation and assessed for normality using the D’Agostino & Pearson omnibus K2 test. The HR, RR, SAP, DAP and MAP data were longitudinally analysed using the Repeated Measures ANOVA and the Holm–Sidak post hoc tests. The RT data were not normally distributed; accordingly, a non-parametric approach was followed using the Friedman test and subsequent Dunn’s multiple comparison test. The depth of sedation, analgesia and ataxia scores were presented as mean ± standard error of the mean and longitudinally analysed using the Friedman test and the Dunn’s multiple comparison test. The frequency analysis for all sequential time points, considering the mules requiring supplemental xylazine or infiltration of the ovarian pedicle with lidocaine, was performed using the Chi-square (χ^2^) test for trend. All data were analysed using the software GraphPad Prism 8 for MacOS, Version 8.2.1 (GraphPad Software Inc., San Diego, CA, USA).

## 3. Results

The sample of eight animals was considered adequate to carry out the present study with interesting results.

Bilateral laparoscopic ovariectomy was safely completed in all eight mules. During pneumoperitoneum, no animals showed discomfort. The mean surgery time measured from first incision to the skin closure of the second flank approach was 106 ± 5 (98–110) min. The procedure was performed by the same two expert surgeons.

There was no significant difference between animals for age, weight and duration of surgery.

The sedation with IM 0.05 mg/kg acepromazine followed by 1.3 mg/kg of xylazine and 0.02 mg/kg of butorphanol intravenously was considered suitable in 5 out of 8 animals. Three animals needed an extra dose of IV 0.3 mg/kg xylazine. The inadequate sedation was observed in one mule that was particularly nervous during the IV catheter placement. In one mule, the first intercoccygeal vertebrae space was more difficult to palpate, because the animal was a slightly obese and required a longer time and one animal tried to jump out the stocks during epidural catheter placement. No animals reported signs of discomfort during pneumoperitoneum induction at carbon dioxide insufflation pressure of 15 mmHg.

Epidural anaesthesia allowed ovarian manipulation in six mules while in two mules the manipulation of the ovary was not completely tolerated. In one mule a supplemental IV dose of xylazine 0.65 mg/kg was sufficient to complete the surgery, while in one mule it was no sufficient and infiltration of the ovarian pedicle with 20 mL of lidocaine 2% was needed to complete the surgery. However, no significant differences were observed considering the frequency trend of mules requiring supplemental xylazine or lidocaine infiltration (χ^2^ = 1.493, *p* = 0.2217).

Mean values ± standard error of depth of sedation, ataxia and analgesia scores during anaesthesia and surgery are reported in Figure 1. The mean values of invasive SAP, DAP, MAP, HR, RR, and RT are reported in Table 3. Arterial blood pressures decreased during the early stages of surgery, then they tended to stabilize in the second half (Table 3 and Figure 2).

The values of HR and RR did not show any statistical difference during the procedure. The RT decreased slightly during surgery with a significant difference between the beginning and the end of the measurements (Table 3 and Figure 3).

Although incisional swelling and gas crepitus may develop at the laparotomy site [4], only one mule developed seroma formation followed by surgical wound dehiscence. The surgical site was treated with daily topical wound debridement and lavage using saline solution and clorhexidine (diluted to 0.05%).

## 4. Discussion

Previous similar studies on mules and donkeys considered adequate a sample of six animals [22,26,27]; therefore, a total of 8 was used to carry out the present study with relevant and encouraging results, although in a small group of animals undergoing surgery. As performed in a previous study, the method used to evaluate the weight of the mules was extrapolated from horses and therefore may not be entirely appropriate for use in this species [21]. The authors preferred this method to the Donkey Sanctuary’s weight estimator [28], considering that mules would be more similar to horses compare to donkeys, whose morphology is really highly variable.

The 1.3 mg/kg dose of IV xylazine was adequate in five out of the eight animals. The insufficient sedation was found in one mule that received premedication IV before the catheter placement. As a result of the nervous behaviour, more time it was required for the catheter placement resulting in a reduction in the effect of the xylazine, as the half-life of this drug is 15 min shorter (32 min) in mules than in horses (47 min) [29].This is in accordance with a previous study, demonstrating that mules may require approximately 50% more xylazine for sedation (and probably other alpha 2 agonists) than either donkeys or horses [1,30]. One mule tried to jump out of the stock during needle epidural placement, and this event may be a possible consequence during the procedure. This event could be a big problem; therefore, during the epidural anaesthesia, the operator should pay attention to confine the animals in a dedicated room. One mule was a slightly obese and required a longer time to identify the first intercoccygeal vertebrae space.

It is known that the degree of sedation and analgesia achieved is dose dependent with sedation lasting longer than analgesia in donkeys after IV alpha2 [30]. In this study, analgesia was greater and lasted equal to or longer than sedation, highlighting the peculiarity of the anaesthetic protocol used which included epidural anaesthesia that seems able to implement and extend analgesia. Although standing laparoscopic ovariectomy is a mini-invasive in nature, it is a painful procedure, especially when the pneumoperitoneum was induced and when the ovarian pedicle is grasped and ligated. For these reasons, a slow carbon dioxide insufflation and an injection of local anaesthetics in the ligament, mesovarium and mesosalpinx or topical anaesthesia are recommended [4,12,31,32,33,34]. In this study, epidural anaesthesia allowed ovarian manipulation in six out of eight mules. In one mule, a supplemental IV dose of xylazine 0.65 mg/kg was sufficient to complete the surgery, while in one mule it was not sufficient, and infiltration of the ovarian pedicle with 20 mL of lidocaine 2% was needed to complete the surgery.

Few papers describe epidural anaesthesia performed during laparoscopic ovariectomy in horses and none in mules [4,35,36,37,38,39,40]. In the present study, the doses used to perform epidural anaesthesia have been extrapolated from previous publications carried out on the horse [37,38,39]. Skarda and Muir [39] reported that epidural administration of detomidine at a dose of 60 μg/kg in a total volume of 10 mL induced in horse variable bilateral analgesia ranging from the first coccygeal to the third sacral and from the first coccygeal to the forteenth thoracic dermatome areas. The same authors reported in another study that caudal epidurally administered xylazine (0.25 mg/kg in 6 mL of 0.9% NaCl) can be given safely to induce prolonged (>2 h) caudal analgesia with minimal sedation, ataxia, and circulatory and respiratory disturbances in conscious, standing mares [40].

Detailed information regarding the exact sensory tracts of the ovary in horses is limited, but Sach and Habel reported that the ovarian nerve plexus enters the caudal mesenteric ganglion, which is located ventral to the lumbar spinal vertebrae 3 (L3) [41]. In this study, epidural anaesthesia allowed ovarian manipulation rather than instilling local anaesthetic into the ovarian pedicle in six out of eight mules, while in two, the manipulation of one ovary was not completely tolerated. Therefore, one mule received a supplemental IV dose of xylazine 0.65 mg/kg to complete the surgery, while the other one the xylazine bolus was no sufficient and the ovarian pedicle was infiltrated with 20 mL of lidocaine 2% to complete the surgery. However, no significant differences were found considering the frequency trend of mules requiring supplemental xylazine or lidocaine infiltration during surgery and therefore it was considered not statistically relevant for this study.

It is important to highlight that the body length of the two mules that received supplementary ovarian local analgesia was longer than the others, which may have impeded the epidural analgesics from reaching the sacral spinal nerve 3 (S3). This hypothesis could lead to conclude that further comparative studies are needed to determine whether the dose of agents administered via the epidural route according to the desired effect should take into account the body weight of the animal or the extension of the vertebral column, as it has been reported in small animals [42,43,44]. Therefore, operators should pay attention more to that physical parameter rather than to the effective weight in mules, which is the main parameter we normally consider when preparing the dose and volume for epidural analgesic drugs combination. In fact, probably in these two mules the intended meaning has been not reached.

Petrizzi et al. successfully carried out bilateral laparoscopic ovariectomies in ten mules performing the infiltration of the ovarian pedicle using 10–20 mL of 2% lidocaine through a laparoscopic needle in order to attain local analgesia before the ovarian removal [5]. However, Farstvedt and co-authors, asserted that neither a lidocaine intraovarian injection nor a mesovarian injection eliminated all pain responses during laparoscopic ovariectomy in mares [33]. In this study, two animals showed lack of analgesia following caudal epidural anaesthesia, implying that probably a combination of both techniques could be the best solution to obtain ovary analgesia during ovariectomy.

In this study, two different scoring tools were used. The first one was applied to evaluate only the degree of sedation immediately after the acepromazine, xylazine and butorphanol administration. The second one was used to evaluate sedation, ataxia and analgesia after epidural anaesthesia. Both scoring systems were on a 4-poit scale avoiding confusion in the evaluation of the results. Although all mules were ataxic at 10 min post-epidural treatment (Figure 1), it was not necessary to stop the CRI of xylazine, because postural instability or swaying with hind limbs crossed were not observed.

The main limitation of this research was that a control group was not included for comparison with the protocol used, because insufficient numbers of mule mares were available to be enrolled in this study.

## 5. Conclusions

In this study, eight healthy adult mule mares sedated with acepromazine, xylazine and butorphanol were submitted to bilateral laparoscopic ovariectomy. The sedation was maintained with by CRI of xylazine which allowed to facilitate standing surgery keeping stable the degree of sedation without any side effects. Before the start of surgery, epidural anaesthesia was performed with 0.2 mg/kg of lidocaine and 0.17 mg/kg of xylazine diluted in 10 mL saline 0.9% and it was injected between Co1 and Co2. The analgesia obtained was insufficient in two mules that requested an additional treatment to complete the surgery. Additional studies with a greater number of animals are needed to determine whether xylazine-lidocaine epidural anaesthesia may have a role in the management of intraoperative pain and whether this technique combined with ovarian pedicle analgesia could be the best solution to perform ovary analgesia during standing laparoscopic ovariectomy.

## Figures and Tables

**Figure 1 animals-11-02419-f001:**
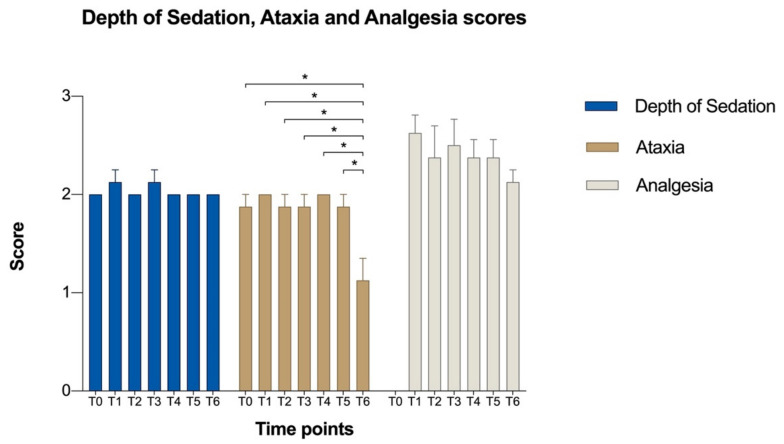
Mean values ± standard error of Depth of Sedation, Ataxia and Analgesia scores during surgery. Time points, T0: 10 min after epidural anaesthesia; T1: immediately after the start of surgery; T2: 15 min, T3: 30 min, T4: 60 min, T5: 90 min at the after the start of surgery; T6: end of surgery. Asterisks indicate significant the differences showed at multiple comparisons between time points, *: *p* < 0.05.

**Figure 2 animals-11-02419-f002:**
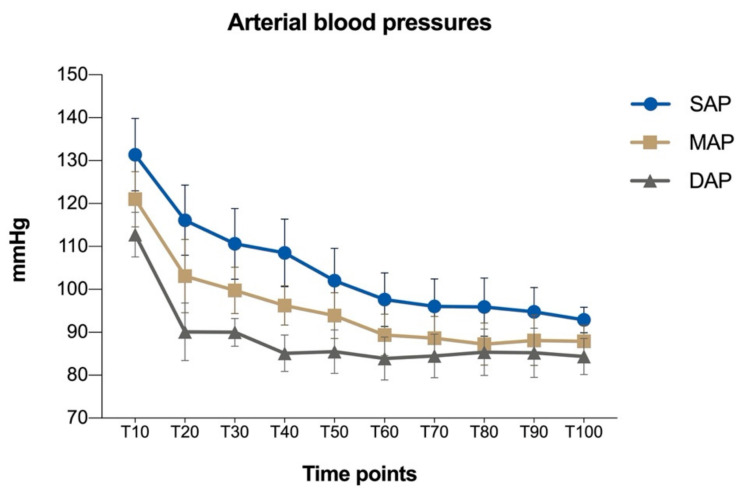
Mean values ± standard deviation of invasive systolic (SAP), mean (MAP) and diastolic (DAP) arterial blood pressures (mmHg) during the study. Time points, T0: baseline; from T10 to T100: measurements from the start of the surgery until the end of the surgery.

**Figure 3 animals-11-02419-f003:**
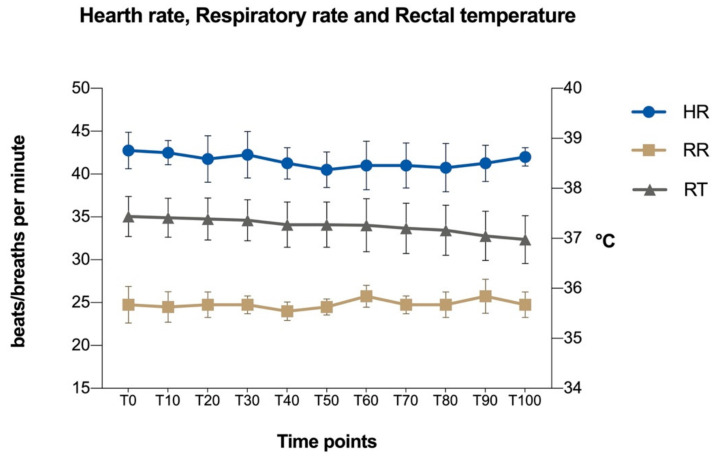
Mean values ± standard deviation of heart rate (HR, beats/min), respiratory rate (RR, breaths/min) and rectal temperature (RT, °C) during the study. Time points, T0: baseline; from T10 to T100: measurements from the start of the surgery until the end of the surgery.

**Table 1 animals-11-02419-t001:** Scoring system used to assess depth of sedation in 8 mules before ovariectomy (from Vullo et al., 2017 [22]).

Scores	Definition of Scoring
Sedation 0 (poor)	Fully responsive to environment, lips apposed, no lowering of head, no drooping of the ears
1 (mild)	Still responsive to environment, slight separation of lower lip, slight lowering of the head, slight drooping of the ears
2 (good)	No response to environment, separation of the lower lip, lowering of the head, drooping of the ears
3 (heavy)	No response to environment, extreme lip separation, pronounced loss of postural tone and ataxia, pronounced separation of the ear tips

**Table 2 animals-11-02419-t002:** Scoring system used to score analgesia, depth of sedation, and ataxia in 8 mules undergoing ovariectomy under acepromazine, xylazine and butorphanol sedation followed by a CRI of xylazine combined with epidural anaesthesia with xylazine-lidocaine [modified from Sampaio et al., 2017 [23] and Schauvliege et al., 2019 [24].

Scores	Definition of Scoring
Analgesic score	
0	Normal response, with a strong and fast reaction to surgical stimulus
1	Mild analgesia, with no immediate response to surgical stimulus with tail swishing, low whole-body reaction, and turning toward the site of painful stimulus
2	Moderate analgesia, with no tail swishing, no response to surgical stimulus, without whole-body reaction but restless
3	Complete analgesia, animal was calm and indifferent to surgical stimulus
Depth of Sedation	
0	No sedation. Animal is alert with normal posture and response to environment. Normal objection to intervention. Ears responsive to surroundings (moving).
1	Mild sedation. May or may not lean on head support, relaxed facial muscles. Reduced responses to background activity in the room. Ears partially responsive to surroundings. Light or no ptosis of the ears.
2	Good sedation. Leans on head support. No response to background activity in the room. Pendulous lower lip. Ears mildly responsive to surroundings. Moderate ear ptosis. Eyelids partially closed.
3	Marked sedation. Leans strongly on head support. No response to background activity in the room. Pendulous lower lip. Pronounced ear ptosis, minimal/no movement of ears. Eyelids partially or fully closed. Eye may be rotated, little to no movements of the eye.
Ataxia	
0	Standing square, bearing equal weight on all four legs.
1	One hind limb in resting position and/or slight swaying.
2	Clear swaying or leaning against the stock (not bearing weight on at most one of the four limbs).
3	Very pronounced leaning (possibly not bearing weight on more than one limb) and/or attempts to become recumbent.

**Table 3 animals-11-02419-t003:** Mean values ± standard deviation and longitudinal statistical results of invasive systolic (SAP), mean (MAP) and diastolic (DAP) arterial blood pressures (mmHg), heart rate (HR, beats/min), respiratory rate (RR, breaths/min) and rectal temperature (RT, °C) during the study. Time points, T0: baseline; from T10 to T100: measurements from start of surgery until the end of surgery. nd: not determined. F: result of Anova F statistics (F-ratio); χ^2^_r_: result of Friedman statistics. For each parameter, the pairs of equal letters indicate the pairs of comparison between time points showing significant difference at post hoc multiple comparisons; asterisks indicate the level of significance at multiple comparisons, *: *p* < 0.05; **: *p* < 0.01; ***: *p* < 0.001; ****: *p* < 0.0001.

Item	T0	T10	T20	T30	T40	T50	T60	T70	T80	T90	T100	Statistical Results
SAP	nd	131 ± 8 ^a *, b *, c **, d **, e **, f **, g **^	116 ± 8 ^h **, i **, j **, k *, l *, m **^	111 ± 8 ^n *, o **, p *, q *, r *, s **^	108 ± 8 ^a *, t *, u **, v **, w *, x *, y *^	102 ± 8 ^b *, h **, n *, t *, z *^	98 ± 6 ^c **, i **, o **, u **, z *^	96 ± 6 ^d **, j **, p *, v **^	96 ± 7 ^e **, k *, q *, w *^	95 ± 6 ^f **, l *, r *, x *^	93 ± 3 ^g **, m **, s **, y *^	F = 36.75 *p* < 0.0001
MAP	nd	121 ± 6 ^a *, b **, c **, d **, e **, f **, g **, h **, i **^	103 ± 8 ^a *, j *, k **, l **, m *, n *^	100 ± 5 ^b **, o **, p **, q **, r *, s *^	96 ± 5 ^c **, t **, u **, v **, w *, x *^	94 ± 5 ^d **, y *, z ****, za **, zb *^	89 ± 5 ^e **, j *, o **, t **, y *^	89 ± 5 ^f **, k **, p **, u **, z ****^	87 ± 5 ^g **, l **, q **, v **, za **^	88 ± 6 ^h **, m *, r *, w *, zb *^	88 ± 4 ^i **, n *, s *, x *^	F = 46.95 *p* < 0.0001
DAP	nd	113 ± 5 ^a *, b **, c **, d **, e ***, f ***, g **, h **, i **^	90 ± 7 ^a *^	90 ± 3 ^b **^	85 ± 4 ^c **^	85 ± 5 ^d **^	84 ± 5 ^e ***^	84 ± 5 ^f ***^	85 ± 5 ^g **^	85 ± 6 ^h **^	84 ± 4 ^i **^	F = 32.74 *p* < 0.0001
HR	43 ± 2	42 ± 1	42 ± 3	42 ± 3	41 ± 2	40 ± 2	41 ± 3	41 ± 3	41 ± 3	41 ± 2	42 ± 1	F = 2.794 *p* = 0.0662
RR	25 ± 2	24 ± 2	25 ± 1	25 ± 1	24 ± 1	24 ± 1	26 ± 1	25 ± 1	25 ± 1	26 ± 2	25 ± 1	F = 1.115 *p* = 0.3647
RT	37.44 ± 0.40 ^a *, b **, c ***^	37.41 ± 0.39 ^d **, e **^	37.39 ± 0.42 ^f *, g **^	37.36 ± 0.41 ^h *^	37.28 ± 0.45	37.28 ± 0.45	37.26 ± 0.53	37.20 ± 0.50	37.16 ± 0.50 ^a *^	37.05 ± 0.49 ^b **, d **, f *^	36.98 ± 0.48 ^c ***, e **, g **, h *^	χ^2^_r_ = 50.05 *p* < 0.0001

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
