# Peer review of "Analgesic and Sedative Effects of Epidural Lidocaine-Xylazine in Elective Bilateral Laparoscopic Ovariectomy in Standing Mule Mares"

_animals, 2021, doi:10.3390/ani11082419_

Round 1

Reviewer 1 Report

Dear Editor and authors

The paper discusses the use of lidocaine-xylazine epidural anesthesia in bilateral laparoscopic ovariectomy in standing mule

The paper is well written, the description very clear, and the results of the sample group were concise. However, the results cannot be conclusive given the lack of a control group.

Mule are hybrid animals (Equus cabalus female with an Equus asinus Male) and have a different physiology from their parents. But I have doubts about be the impact of the paper, since there are similar papers that use the same drugs in close doses in Equus cabalus that were not mentioned.

Grubb, T. L., T. W. Riebold, and M. J. Huber. "Comparison of lidocaine, xylazine, and xylazine/lidocaine for caudal epidural analgesia in horses." Journal of the American Veterinary Medical Association 201.8 (1992): 1187-1190.

LeBlanc, P. H., et al. "Epidural injection of xylazine for perineal analgesia in horses." Journal of the American Veterinary Medical Association 193.11 (1988): 1405-1408.

LeBlanc, P. H., and J. P. Caron. "Clinical use of epidural xylazine in the horse." Equine Veterinary Journal 22.3 (1990): 180-181.

I recommend that the authors include the control group as soon as possible, and also note whether the statement by Latzel (2008) who reported the pharmacokinetics of xylazine in mules compared to horses that recommended a 50% higher dose in mules is true or not 

Latzel, S.T. (2008) Clinical and Pharmacological Studies on Elimination Kinetics of Xylazine (RompunR/Bayer) in Mules. Dissertation (thesis), Ludwig-Maximillians-University, Faculty of Veterinary Medicine, Munich, 1st edn., Verlag Dr Hut, Munich. ISBN 978-3-89963-894-3.

Author Response

thank you very much for your constructive comments. As you suggested, we have tried to improve the quality of the manuscript accordingly.

As requested, all changes made to the manuscript, in comparison to the previous version, have been reported in the text using “track changes” in Microsoft Word.

Below, the answers of the authors to the comments.

Based on the recent publications mentioned in the paper, we hope that this work has the potential to gain good visibility, despite the fact that similar work has already been published. In fact, there has been growing interest in the mule, about which little information has yet to be published in the field of anaesthesiology.

We have added in the text the important citations that you suggested.

Despite the fact that we considered the control group to be important in order to compare the results, we were not able to include it since, as we affirmed in the conclusion, it was very difficult to find sufficient animals to do so.

Reviewer 2 Report

Dear Authors, thank you for allowing me to review this paper which describes the use of epidural lidocaine-xylazine to provide analgesia in mules undergoing bilateral standing ovariectomy.

I found this paper interesting and essentially written in a comprehensive English, however some corrections and better address of some details are required before final approval.

Simple summary:

LL.13-14:  ‘…to correct undesirable behavior effects related to estrus and also to improve…’

  1. 15: ‘…in standing position with a combination of sedative and analgesic drugs along with local anaesthesia.’ you might also need desensitization of skin and subcutaneous layers beside ovarian pedicle.
  2. 18-19: I would remove ‘in standing position’. According to what commented later on, please verify your assumption for ‘baseline values’ as it is not completely clear if this is before any given drug or 10’ after epidural

LL 20-21. ‘Epidural analgesia with lidocaine-xylazine may be a good alternative to local anaesthetics infiltration of the ovarian pedicle for…’, if instillation means ‘splash block’ I would give a wider possibility of application of local anaesthetics.

Abstract:

L.24 ‘…lidocaine-xylazine administration…’

L.32 ‘…successfully completed in all animals’.

  1. 34 ‘…out of eight animals’. Please reformulate the sentence with ‘painless’ expression, as animals did not show any signs of discomfort associated with nociception … it’s probably better.

Or ‘Animals were mostly calm and with adequate level of analgesia during the procedures, however..’

Keywords

Ataxia was not addressed in your results and discussion as commented later on although significant values were recorded

Introduction

  1. 41. probably ‘Mule’ at singular is better (then hybrid and combines)

L.47: ‘…., as in equine mares,’

  1. 48: ‘…to decrease visceral pain…’ rather than for the abolition
  2. 49: you may write [4-6] when you have consecutive numbers of bibliography, similar at Line 66 [8-10], line 71 [13-15]; L 76 [18-20]

  1. 50: rather, ‘ …by laparotomy, under general anaesthesia, or by laparoscopy which can be performed either in the standing sedated horse or in dorsal recumbency under general anaesthesia’

LL 51-54. Somewhere I would include also the lower costs compared to general anaesthesia.

  1. 55-56 The response to surgical stimulus is rather linked to the level of analgesia, I suppose, therefore I would post-pone the ataxia afterword.

  1. 59 ‘..and AT the 17th intercostal…’
  2. 60-62. Are you sure that this description is intended for an inverted L-block? Could you please provide a reference for that? I believe that this is basically the description of infiltration anaesthesia of portal sites for trocars. Otherwise, this technique would be more appropriately described as an injection of the tissues you’ve mentioned but bordering the dorso-caudal aspect of the last rib and the ventro-lateral aspect of the lumbar transverse processes.

Your ref. n°12 describes comparison between infiltration or splash (i.e. simple topical instillation) of local anaesthetics, with the latter technique probably aiming at reducing the little trauma caused by the needle (which in turn can cause hemorrhage thereafter), please try to reformulate.

Coccygeal vertebrae are reported as Co, while C is intended for cervical, please amend where required.

  1. 65-66 it is not only easier or safer in the field, but everywhere. ‘… and reduces the potential for nerve damage with respect to subarachnoid anaesthesia.’ Please reformulate properly, you never mentioned subarachnoid anaesthesia. Indeed, with the caudal approach you may reduce the risk of inadvertent injection in the subarachnoid space as well as the risk of motor blockade of the pelvic limbs, compared to the cranial approach. However, the cauda equina could still be injured. I would also say ‘Caudal epidural anaesthesia is used to desensitize the coccygeal and last two or three sacral nerves, …’.

L 72-76 ref. n13 is not appropriate at this point as it does not refer to surgeries with flank approach, while n.14,15 do. Description of the ligamentum flavum is also appropriated for n.16.

Ref n.14 does not describe complications related to caudal epidural while n.16,17 do.

LL 74-76. ‘…motor and sympathetic blockage’. Please try to rephrase the following sentence. Period of action could be simply the duration.

M&M

  1. be consistent in the way you write your ASA status: as later on you write ASA >2 write, 1 at this point rather than I,
  2. 89. Use first the complete definition for your IM abbreviation

LL.92- 93; ‘…estimated on the basis of…’ was any physical restraint provided as well? i.e. tail rope, headcollar ropes? head support? ‘..were placed in a stock’, without standing.

LL 94-95: this sentence has poor significance, as there was no need for blindness, sedation could have been performed by anyone, indeed a description on the person who performed scoring during the procedures would be more indicated, i.e. was this always the same person? was the anesthetist who performed also the epidural injection? furthermore, why field surgery? could you describe more accurately your environment? was there any eventual disturbing factors for animals which could have contributed to a sudden recovery from deep sedation?

  1. 95-96: was baseline measured before sedation? please verify this assumption as later on you describe differently. When you describe your parameters in parenthesis you should mention the abbreviation and your measure units: therefore (RT, °C); (HR, beats/min); (RR, breaths/min). I would move temperature at the end, given its lower relevance, and the description should be modified accordingly.

L.100-102 here you can use IM and IV as you already mentioned the abbreviation at LL. 89-90.

LL 103-104: when was the additional xylazine of 0.3 mg/kg administered? was it during surgery or only before epidural administration? How often was the scale in table 1 applied and when for the first time? however, later on you show a different sedation score, why did you use two different tools? this should be addressed somewhere in your discussion and/or results. Moreover, please refer to any reference if this scale (table 1) is from previous studies.

LL 107-108: ‘…to minimize visual and auditory stimulation, respectively.’

LL 109-110: what kind of infiltration did you perform? ‘Each paralumbar fossa was infiltrated with....’  3 portal sites? Somewhere you should add a description of your surgical condition and environment...as the earlier term 'field' could confound the reader otherwise. Were the animals sedated when already in the stock or before entering the stock? did you secure tails and head to ropes anywhere? or they were freely movable? Please describe

LL 110-115: was any local anesthetic bleb for skin and subcutaneous layers above the epidural space provided before your quincke needle introduction? Your spinal needle would be 18 gauge size,‘…at 30-45 degree angle intothe specified space’. Was LOR tested with air before or directly with anaesthetics?

LL 118-119: I did not get this final procedure, please explain better. Which needle (why plural?) did you fill up with 0.5 ml of saline? the spinal needle? did you simply fill the hub or you injected further 0.5 ml then? please try to explain better

  1. 119: ‘..laparoscopic ovariectomy started 15 minutes after drug administration’ do you mean after epidural? please describe a bit your surgical procedures at least in terms of number of portal access (were your 30 mls of lidocaine intended for this use and this specific location?) moreover, as later on you mention that you used capnoperitoneum during your procedures you should describe as well, as this is reported as a possible discomfort due to peritoneal irritation.

LL 120-122: please describe the artery chosen for your invasive measurement, however, earlier you stated that you would auscultate for heart rate, is that still correct? or did you use the monitor at this point? try to rephrase this sentence, i.e. ‘Invasive blood pressure was continuously monitored through a multiparametric monitor (which further allowed for rectal temperature and ECG recording ?and therefore RT and HR monitoring’...if this is the case...). As far as body temperature, you first used RT as abbreviation for this parameter, Celsius would only be your unit of measure- please amend and be consistent.

  1. 122-123: ‘ ‘Depth of sedation and ataxia were scored 10 mins after the epidural anaesthesia (T0)’. And immediately following you state ‘Analgesia, depth of sedation…’ are those evaluations different? please explain better as it is not clear. Probably is only analgesia which starts to be scored at the beginning of surgery. However, this could be the reason for your two different scoring tools, but it is not clear enough in my opinion, moreover, as they are both on a 4-point scale I rather would have used the same to limit confusion. In any case, please try to explain better and this should be discussed as well.
  2. 123-126: You start your sentence indicating that you would score analgesia, sedation and ataxia and I would place your (Table 2) right after your references. At L 125: I would remove ‘…from the beginning of the surgery’ and simply state ‘..thereafter and again at the end of surgery (T6)’; But there’s no need to repeat ‘At T1, T2,T3, T4, T5, and T6 analgesia was also evaluated’ if you can describe earlier in a different way, please remove (LL126-127).

L 127 ‘If the quality of sedation was…’. Was supplemental xylazine administered if sed. score was 0 as you reported at line 103, or also analgesia score played a role? please specify which score did you consider inadequate.

  1. 130: which dose of 2% lidocaine was infiltrated into the ovarian pedicle?

  1. 131: Table 2. Your statement ‘…under xylazine sedation…’ is not correct as you also used acepromazine and butorphanol, and xylazine CRI, please add.

According to your Score for sedation, it seems you had descriptors which should be defined earlier: head support and that the procedure was performed in a room not in the field.

How could you score for sedation depth using eyelids closure item if eyes were covered with a drape (L. 107)? Did you remove the drape for this purpose?

L.145: please include here how your data will be presented, i.e. as mean ± SD, and eventually (range), and mean ±SE for categorical data

Results:

Please rearrange you order in presenting data which should reflect the way you evaluated them:

  • Describe the adequacy of your group of animals (LL 151-152)

  • Then 147-150. I would say ‘Bilateral standing laparoscopic ovariectomy was safely completed in all 8 mules’. Is skin closure intended of the second flank approach? Please include. please move your measure unit at the end of your numbers. i.e. 106±5 (98-110) minutes (I would round your decimals, here and especially later on when you describe your physiological parameters).

  • Then the adequacy of your sedation (L 165-167) again, amend at line 165 as you did not inject only xylazine, you had acp, butorphanol and a xyl CRI running already (I believe?) I therefore would only mention that sedation was not adequate (for performing epidural??) in 6/8? animals? wasn't your total number of 8 mules? Why do you say ‘in 5 out of 6’…? what about the other 2 animals? the 2 animals that you described behaving differently (one jumping out during epidural and the other being obese), did they require extra xylazine? otherwise it is not clear your sentence here. 5 animals were OK with the first sedative combination, 3 animals probably needed an extra dose of 0.3 mg/kg Xyl IV at a defined time: 1) during epidural needle placement as it was obese and required a longer time; 2) during epidural administration as it tended to jump out of the stock; 3) ... no description.

  • Your parameters. LL 153-164: remove /min from HR and RR as well your unit from RT, as you already described them and later on in the table you'll present again

‘…with a statistical difference..’ replace with ‘significant difference, or statistically significant difference’,

‘…during the procedure’ rather than ‘the study’;

could you better define these ‘early and late time-points’ as we know of intervals of 10minutes from beginning of surgery.

Table 3: L 160: your measure for HR is beats/min, RR breaths/min and BP values in mmHg. As far as you timepoints, is T0 basal before ANY given drug or before surgery? Why do you define before anaesthesia? at line 123 and later on at line 184 you describe this time as 10' after epidural, please clarify and amend where required.

Please use only two decimals for T°, and none for the other physiological parameters' mean values, as HR of 42.75 means simply 43 and SD is 2.

The baseline values seem missing in this table, furthermore, I believe that a graph would be better to describe your results. I would further describe your results in order of clinical importance and especially in order of your first mention in your M&M, i.e. rectal temperature would probably not be the first in my opinion, for its lower relevance when monitoring clinical animals.

  • L 168-169 You refer only here about capnoperitoneum, Please mention earlier that you used it for all your surgeries, if this is the case.

  • Adequacy of epidural (L 173-181): at line 173 you mention 2 mules where inadequate sedation/analgesia was evident, were these 2 the same mules which did not result sedated during epidural? Any comment on that? L. 176: was this infiltration of lidocaine required for one ovary only or for both? was it the first or the second ovary?

Figure 1. in your caption I would remove the term ‘anAesthesia’ if this is only the sore that you used during surgery. Try to use the same unit along the text, therefore min in place of minutes. L185: remove THE from ‘start of surgery’; T6 ‘at the end of surgery’

  • Your post-op complications: Lines 170-172. ‘…gas crepitus may develop…’ is this a report from literature (in case you’ll need to add ref) or is this your result? in the latter you need to reformulate; ‘…at the one laparotomy site..’ which laparotomy site? you described only laparoscopy...indeed you might describe (as suggested earlier) a bit more in details your laparoscopic portals as well as the further need to connect two of them (I believe) to make a larger skin incision in order to remove the ovary (which probably could transiently develop gas crepitus and swelling...if I get it properly!)

Discussion

Authors did not discuss neither present as results, their ataxia scores which had significant differences during time, as reported in figure 1.

Moreover it seems that analgesia is greater and lasting equally or longer than sedation which is normally not seen when alpha2 (and opioids) are used in horses or donkeys too. It is known that the degree of sedation and analgesia achieved is dose dependent with sedation lasting longer than analgesia in donkeys after IV alpha2 (Lizarraga et al. Vet Anaesth Analg 2017;44:509–17.) maybe a comment on this would add something to your paper highlighting the peculiarity of your anaesthetic protocol which seems able to implement and extend analgesia.

LL.189-190. I would reformulate, rather than ‘…with significant results’, probably it would be better to define as ‘interesting/relevant and encouraging results’ although in a small group of animals undergoing surgery. I believe that it is more correct, given the low number of animals (which is indeed low even in your reported references that moreover were not clinical cases i.e. animals did not undergo surgery but only evaluation of effects of different protocols).

LL.191-193. In the authors knowledge, are there any nomogram similar to that developed for donkeys to measure weight differently from horses that could be applied for mules? However, probably mules would be more similar to horses compare to donkeys, whose morphology is really highly variable.

  1. 194 ‘..in seven out of eight animals.’ earlier (line 165) you describe in 5 out of 6..., but also you described two mules requiring extra xylazine for epidural….please amend where required.
  2. 194-197. this part is very similar to your study n.26; however, 1 mule is not as significant in leading to your conclusion of the inadequate dose. Moreover, studies showing the requirement of approximately 50% more xylazine (1.6 mg/kg IV) or detomidine (0.03 mg/kg IV) to produce good sedation than most donkeys or horses, come from Matthews and coworkers, please include this author as reference for this statement (i.e. Matthews et al. Anaesthesia of donkeys and mules. Equine vet. Educ. (1997)) as in study n.26 the same dose of 1.3 had been used with again only 1 animal/8 showing inadequate level of sedation.

Your assumption ‘The inadequate sedation was observed in one mule…’ is that referred to the epidural procedure? If this mule was nervous, would it not be excluded from the study given your assumption at LL.86-87 …. Was this mule not considered as intractable behaviour? ‘…. More time was required…’ remove IT; is there a mean reportable time that could be shown to better address this issue? was this the same mule that later on required further drugs also for the surgical procedure?

L 202. Please reformulate the sentence with ‘…possible effect…’, moreover your reference is not appropriate, as this is a review for equine standing laparoscopy in equine but there’s no real mention to possible side effects at the time of epidural placement, while the ‘jumping out’ is referred to skin clamp placement during draping. Did this mule receive the extra xylazine as well although its sedation was probably adequate according to scoring system of table 1?

  1. 204. ‘…is a painful procedure,…’ I would rather say that, ‘…although mini-invasive in nature, it is a painful procedure, especially when...’ '..is grasped and ligated'....which technique did the authors used for ovariectomy?, ‘…or simply topical anaesthesia’ [12,28-30].

  1. 207: is ‘Very few papers…’ referred to n.31 and n.32?? please include them after ‘…in mules’.

  1. 218 I would enter the paragraph which start with ‘In this study,…’ and say ‘In the present study,…’

  1. 220-223. ‘…without infiltrating local…’ rather than instilling, however, at lines 174-177 you described that only 1 mule required 20 ml lidocaine into the ovarian pedicle, here they are 2, please amend where required. ‘…while in two the manipulation….’ are those other 2 mules? or it is the same concept?

221-223. This concept is very interesting but not very clear, you mention that the body length was greater (weren't mules similar in weight as you stated in your results?) was the body weight similar instead? So that operators should pay attention more to that physical parameter rather than to the effective weight in mules, which is the main parameter we normally consider when preparing the dose and volume for epidural combinations, ‘..epidural analgesic drugs combination…’

However, I wonder how L3 could be reached with your caudal epidural approach with only 10 ml volume. Please verify your assumption, as normally maximal dermatomal analgesic spread ranges from C1-S3 spinal cord segments with caudal epidural injection.

  1. 226: ‘…vertebral column, as IT has been reported… [34-36]’.

  1. 233: ‘In this study, two animals also showed lack of analgesia following …’ why also?

Conclusion

This part could be improved, moreover, don’t forget to mention that your epidural was combined with a continuous rate infusion of xylazine (which has not been so largely described in mules)

Please amend abbreviation for Coccygeal vertebrae (Co)

References

In general, there are some typos in your references.

n.1 Matthews, then a comma

n.5 check the name of the second author

n.6 check the name of the first author

n.7 check adequacy for journal

n.9-10 Comma after the surname

n.11 check names

n.14 check names

n.17 check format for journal writing

  1. 19 check name second author

n.20 check format

n.22 comma after DeRossi and full stop after Sci

n.23 space after 2019

n.27 Latzel,

n.28 check format

n.30 check (lidocaine in)

n.32 mares; check reference details for Volume and format

n.34 verify adequacy from guidelines

n.35 check writing format

n.36 check writing format

Author Response

Dear Authors, thank you for allowing me to review this paper which describes the use of epidural lidocaine-xylazine to provide analgesia in mules undergoing bilateral standing ovariectomy.

I found this paper interesting and essentially written in a comprehensive English, however some corrections and better address of some details are required before final approval.

Thank you very much for your constructive comments. As you suggested, we tried to improve the quality of the manuscript accordingly.

As requested, all changes made to the manuscript, in comparison to the previous version, have been reported in the text using “track changes” in Microsoft word.

Below, the answers of the authors to the comments.

Simple summary:

LL.13-14:  ‘…to correct undesirable behavior effects related to estrus and also to improve…’

  1. 15: ‘…in standing position with a combination of sedative and analgesic drugs along with local anaesthesia.’ you might also need desensitization of skin and subcutaneous layers beside ovarian pedicle.
  2. 18-19: I would remove ‘in standing position’. According to what commented later on, please verify your assumption for ‘baseline values’ as it is not completely clear if this is before any given drug or 10’ after epidural

LL 20-21. ‘Epidural analgesia with lidocaine-xylazine may be a good alternative to local anaesthetics infiltration of the ovarian pedicle for…’, if instillation means ‘splash block’ I would give a wider possibility of application of local anaesthetics.

Modified as suggested.

Abstract:

L.24 ‘…lidocaine-xylazine administration…’

L.32 ‘…successfully completed in all animals’.

  1. 34 ‘…out of eight animals’. Please reformulate the sentence with ‘painless’ expression, as animals did not show any signs of discomfort associated with nociception … it’s probably better.

Or ‘Animals were mostly calm and with adequate level of analgesia during the procedures, however..’

Modified as suggested.

Keywords

Ataxia was not addressed in your results and discussion as commented later on although significant values were recorded

Modified as suggested.

Introduction

  1. 41. probably ‘Mule’ at singular is better (then hybrid and combines)

 L.47: ‘…., as in equine mares,’

  1. 48: ‘…to decrease visceral pain…’ rather than for the abolition
  2. 49: you may write [4-6] when you have consecutive numbers of bibliography, similar at Line 66 [8-10], line 71 [13-15]; L 76 [18-20]
  3. 50: rather, ‘ …by laparotomy, under general anaesthesia, or by laparoscopy which can be performed either in the standing sedated horse or in dorsal recumbency under general anaesthesia’

Modified as suggested.

LL 51-54. Somewhere I would include also the lower costs compared to general anaesthesia.

  1. 55-56 The response to surgical stimulus is rather linked to the level of analgesia, I suppose, therefore I would post-pone the ataxia afterword.

Modified as suggested.

  1. 59 ‘..and AT the 17th intercostal…’
  2. 60-62. Are you sure that this description is intended for an inverted L-block? Could you please provide a reference for that? I believe that this is basically the description of infiltration anaesthesia of portal sites for trocars. Otherwise, this technique would be more appropriately described as an injection of the tissues you’ve mentioned but bordering the dorso-caudal aspect of the last rib and the ventro-lateral aspect of the lumbar transverse processes.

Modified as suggested.

Your ref. n°12 describes comparison between infiltration or splash (i.e. simple topical instillation) of local anaesthetics, with the latter technique probably aiming at reducing the little trauma caused by the needle (which in turn can cause hemorrhage thereafter), please try to reformulate.

Coccygeal vertebrae are reported as Co, while C is intended for cervical, please amend where required.

Modified as suggested.

  1. 65-66 it is not only easier or safer in the field, but everywhere. ‘… and reduces the potential for nerve damage with respect to subarachnoid anaesthesia.’ Please reformulate properly, you never mentioned subarachnoid anaesthesia. Indeed, with the caudal approach you may reduce the risk of inadvertent injection in the subarachnoid space as well as the risk of motor blockade of the pelvic limbs, compared to the cranial approach. However, the cauda equina could still be injured. I would also say ‘Caudal epidural anaesthesia is used to desensitize the coccygeal and last two or three sacral nerves, …’.

Modified as suggested.

L 72-76 ref. n13 is not appropriate at this point as it does not refer to surgeries with flank approach, while n.14,15 do. Description of the ligamentum flavum is also appropriated for n.16.

Ref n.14 does not describe complications related to caudal epidural while n.16,17 do.

Modified as suggested.

LL 74-76. ‘…motor and sympathetic blockage’. Please try to rephrase the following sentence. Period of action could be simply the duration.

Modified as suggested.

M&M

  1. be consistent in the way you write your ASA status: as later on you write ASA >2 write, 1 at this point rather than I,
  2. 89. Use first the complete definition for your IM abbreviation

LL.92- 93; ‘…estimated on the basis of…’ was any physical restraint provided as well? i.e. tail rope, headcollar ropes? head support? ‘..were placed in a stock’, without standing.

Modified as suggested.

LL 94-95: this sentence has poor significance, as there was no need for blindness, sedation could have been performed by anyone, indeed a description on the person who performed scoring during the procedures would be more indicated, i.e. was this always the same person? was the anesthetist who performed also the epidural injection? furthermore, why field surgery? could you describe more accurately your environment? was there any eventual disturbing factors for animals which could have contributed to a sudden recovery from deep sedation?

Further information added.

  1. 95-96: was baseline measured before sedation? please verify this assumption as later on you describe differently. When you describe your parameters in parenthesis you should mention the abbreviation and your measure units: therefore (RT, °C); (HR, beats/min); (RR, breaths/min). I would move temperature at the end, given its lower relevance, and the description should be modified accordingly.

Modified as suggested.

L.100-102 here you can use IM and IV as you already mentioned the abbreviation at LL. 89-90.

Modified as suggested.

LL 103-104: when was the additional xylazine of 0.3 mg/kg administered? was it during surgery or only before epidural administration? How often was the scale in table 1 applied and when for the first time? however, later on you show a different sedation score, why did you use two different tools? this should be addressed somewhere in your discussion and/or results. Moreover, please refer to any reference if this scale (table 1) is from previous studies.

Details requested have been added.

LL 107-108: ‘…to minimize visual and auditory stimulation, respectively.’

LL 109-110: what kind of infiltration did you perform? ‘Each paralumbar fossa was infiltrated with....’  3 portal sites? Somewhere you should add a description of your surgical condition and environment...as the earlier term 'field' could confound the reader otherwise. Were the animals sedated when already in the stock or before entering the stock? did you secure tails and head to ropes anywhere? or they were freely movable? Please describe

Description requested has been added.

LL 110-115: was any local anesthetic bleb for skin and subcutaneous layers above the epidural space provided before your quincke needle introduction? Your spinal needle would be 18 gauge size,‘…at 30-45 degree angle into the specified space’. Was LOR tested with air before or directly with anaesthetics?

Modified as suggested.

LL 118-119: I did not get this final procedure, please explain better. Which needle (why plural?) did you fill up with 0.5 ml of saline? the spinal needle? did you simply fill the hub or you injected further 0.5 ml then? please try to explain better

Better explanation of the procedure added in the text.

  1. 119: ‘..laparoscopic ovariectomy started 15 minutes after drug administration’ do you mean after epidural? please describe a bit your surgical procedures at least in terms of number of portal access (were your 30 mls of lidocaine intended for this use and this specific location?) moreover, as later on you mention that you used capnoperitoneum during your procedures you should describe as well, as this is reported as a possible discomfort due to peritoneal irritation.

Information requested have been added the text.

LL 120-122: please describe the artery chosen for your invasive measurement, however, earlier you stated that you would auscultate for heart rate, is that still correct? or did you use the monitor at this point? try to rephrase this sentence, i.e. ‘Invasive blood pressure was continuously monitored through a multiparametric monitor (which further allowed for rectal temperature and ECG recording ?and therefore RT and HR monitoring’...if this is the case...). As far as body temperature, you first used RT as abbreviation for this parameter, Celsius would only be your unit of measure- please amend and be consistent.

Sentence has been rephrased and changes requested have been made.

  1. 122-123: ‘ ‘Depth of sedation and ataxia were scored 10 mins after the epidural anaesthesia (T0)’. And immediately following you state ‘Analgesia, depth of sedation…’ are those evaluations different?

Yes, those are different evaluations, assessed using two different scores.

  1. please explain better as it is not clear. Probably is only analgesia which starts to be scored at the beginning of surgery. However, this could be the reason for your two different scoring tools, but it is not clear enough in my opinion, moreover, as they are both on a 4-point scale I rather would have used the same to limit confusion. In any case, please try to explain better and this should be discussed as well.

Better explanation of the two scoring tools has been added here and also in the discussion.

  1. 123-126: You start your sentence indicating that you would score analgesia, sedation and ataxia and I would place your (Table 2) right after your references. At L 125: I would remove ‘…from the beginning of the surgery’ and simply state ‘..thereafter and again at the end of surgery (T6)’; But there’s no need to repeat ‘At T1, T2,T3, T4, T5, and T6 analgesia was also evaluated’ if you can describe earlier in a different way, please remove (LL126-127).

Modified as suggested.

L 127 ‘If the quality of sedation was…’. Was supplemental xylazine administered if sed. score was 0 as you reported at line 103, or also analgesia score played a role? please specify which score did you consider inadequate.

Clarification added in the text.

  1. 130: which dose of 2% lidocaine was infiltrated into the ovarian pedicle?

20 ml.

  1. 131: Table 2. Your statement ‘…under xylazine sedation…’ is not correct as you also used acepromazine and butorphanol, and xylazine CRI, please add.

Modified as suggested.

According to your Score for sedation, it seems you had descriptors which should be defined earlier: head support and that the procedure was performed in a room not in the field.

How could you score for sedation depth using eyelids closure item if eyes were covered with a drape (L. 107)? Did you remove the drape for this purpose?

Clarification of the procedure added.

L.145: please include here how your data will be presented, i.e. as mean ± SD, and eventually (range), and mean ±SE for categorical data

The method used to present data was added in the statistical analysis paragraph, at the end of the M&M section.

Results:

Please rearrange you order in presenting data which should reflect the way you evaluated them:

  • Describe the adequacy of your group of animals (LL 151-152)

Description requested added.

  • Then 147-150. I would say ‘Bilateral standing laparoscopic ovariectomy was safely completed in all 8 mules’. Is skin closure intended of the second flank approach? Please include.

Modified as suggested.

  • please move your measure unit at the end of your numbers. i.e. 106±5 (98-110) minutes (I would round your decimals, here and especially later on when you describe your physiological parameters).

Modified as suggested. The decimals have been rounded up here and in the subsequent results (Table 3) (two decimals were used for body temperature and none for the other parameters).

  • Then the adequacy of your sedation (L 165-167) again, amend at line 165 as you did not inject only xylazine, you had acp, butorphanol and a xyl CRI running already (I believe?) I therefore would only mention that sedation was not adequate (for performing epidural??) in 6/8? animals? wasn't your total number of 8 mules? Why do you say ‘in 5 out of 6’…? what about the other 2 animals? the 2 animals that you described behaving differently (one jumping out during epidural and the other being obese), did they require extra xylazine? otherwise it is not clear your sentence here. 5 animals were OK with the first sedative combination, 3 animals probably needed an extra dose of 0.3 mg/kg Xyl IV at a defined time: 1) during epidural needle placement as it was obese and required a longer time; 2) during epidural administration as it tended to jump out of the stock; 3) ... no description.

Added in the manuscript: The sedation with IM0.05 mg/kg acepromazine followed by 1.3 mg/kg of xylazine and 0.02 mg/kg of butorphanol intravenously was adequate in 5 out of 8 animals. Three animals needed an extra dose of IV 0.3 mg/kg xylazine. The inadequate sedation was observed in one mule that was particularly nervous during the IV catheter placement. In one mule, the first intercoccygeal vertebrae space was more difficult to palpate, because the animal was a slightly obese and required a longer time and one animal tried to jump out the stocks during epidural catheter placement. (LL 200-206)

  • Your parameters. LL 153-164: remove /min from HR and RR as well your unit from RT, as you already described them and later on in the table you'll present again

‘…with a statistical difference..’ replace with ‘significant difference, or statistically significant difference’,

‘…during the procedure’ rather than ‘the study’;

could you better define these ‘early and late time-points’ as we know of intervals of 10minutes from beginning of surgery.

Modified as suggested.

Table 3: L 160: your measure for HR is beats/min, RR breaths/min and BP values in mmHg.

Modified as suggested.

 As far as you timepoints, is T0 basal before ANY given drug or before surgery? Why do you define before anaesthesia? at line 123 and later on at line 184 you describe this time as 10' after epidural, please clarify and amend where required.

Clarification added in the text.

Please use only two decimals for T°, and none for the other physiological parameters' mean values, as HR of 42.75 means simply 43 and SD is 2.

Modified as suggested. The decimals were rounded up in Table 3: two decimals were used for body temperature and none for the other parameters.

The baseline values seem missing in this table, furthermore, I believe that a graph would be better to describe your results. I would further describe your results in order of clinical importance and especially in order of your first mention in your M&M, i.e. rectal temperature would probably not be the first in my opinion, for its lower relevance when monitoring clinical animals.

Modified as suggested. Data in Table 3 were presented in order of first mention in M&M.

  • L 168-169 You refer only here about capnoperitoneum, please mention earlier that you used it for all your surgeries, if this is the case.

Indication has been made also earlier in the text.

  • Adequacy of epidural (L 173-181): at line 173 you mention 2 mules where inadequate sedation/analgesia was evident, were these 2 the same mules which did not result sedated during epidural? Any comment on that? L. 176: was this infiltration of lidocaine required for one ovary only or for both? was it the first or the second ovary?

Clarification added in the text.

Figure 1. in your caption I would remove the term ‘anAesthesia’ if this is only the sore that you used during surgery. Try to use the same unit along the text, therefore min in place of minutes. L185: remove THE from ‘start of surgery’; T6 ‘at the end of surgery’

Modified as suggested.

  • Your post-op complications: Lines 170-172. ‘…gas crepitus may develop…’ is this a report from literature (in case you’ll need to add ref) or is this your result? in the latter you need to reformulate; ‘…at the one laparotomy site..’ which laparotomy site? you described only laparoscopy...indeed you might describe (as suggested earlier) a bit more in details your laparoscopic portals as well as the further need to connect two of them (I believe) to make a larger skin incision in order to remove the ovary (which probably could transiently develop gas crepitus and swelling...if I get it properly!)

Details of the laparoscopic procedure have been better described. Post-op complications sentence has been reformulated.

Discussion

Authors did not discuss neither present as results, their ataxia scores which had significant differences during time, as reported in figure 1.

We added in the text: Although all mules were ataxic at 10 min post-epidural treatment (Figure 1), it wasn’t necessary to stop the CRI of xylazine, because postural instability or swaying with hind limbs crossed were not observed.

Moreover it seems that analgesia is greater and lasting equally or longer than sedation which is normally not seen when alpha2 (and opioids) are used in horses or donkeys too. It is known that the degree of sedation and analgesia achieved is dose dependent with sedation lasting longer than analgesia in donkeys after IV alpha2 (Lizarraga et al. Vet Anaesth Analg 2017;44:509–17.) maybe a comment on this would add something to your paper highlighting the peculiarity of your anaesthetic protocol which seems able to implement and extend analgesia.

I added in the text: It is known that the degree of sedation and analgesia achieved is dose dependent with sedation lasting longer than analgesia in donkeys after IV alpha2 [30]. In this study, analgesia was greater and lasting equally or longer than sedation highlighting the peculiarity of the anaesthetic protocol used which included epidural anaesthesia that seems able to implement and extend analgesia.

LL.189-190. I would reformulate, rather than ‘…with significant results’, probably it would be better to define as ‘interesting/relevant and encouraging results’ although in a small group of animals undergoing surgery. I believe that it is more correct, given the low number of animals (which is indeed low even in your reported references that moreover were not clinical cases i.e. animals did not undergo surgery but only evaluation of effects of different protocols).

Modified as suggested.

LL.191-193. In the authors knowledge, are there any nomogram similar to that developed for donkeys to measure weight differently from horses that could be applied for mules? However, probably mules would be more similar to horses compare to donkeys, whose morphology is really highly variable.

Added, as suggested, :“The authors preferred this method to the Donkey Sanctuary’s weight estimator [29], considering that mules would be more similar to horses compare to donkeys, whose morphology is really highly variable”.

  1. 194 ‘..in seven out of eight animals.’ earlier (line 165) you describe in 5 out of 6..., but also you described two mules requiring extra xylazine for epidural….please amend where required.

Correction has been made.

  1. 194-197. this part is very similar to your study n.26; however, 1 mule is not as significant in leading to your conclusion of the inadequate dose. Moreover, studies showing the requirement of approximately 50% more xylazine (1.6 mg/kg IV) or detomidine (0.03 mg/kg IV) to produce good sedation than most donkeys or horses, come from Matthews and coworkers, please include this author as reference for this statement (i.e. Matthews et al. Anaesthesia of donkeys and mules. Equine vet. Educ. (1997)) as in study n.26 the same dose of 1.3 had been used with again only 1 animal/8 showing inadequate level of sedation.

Ok, done.

Your assumption ‘The inadequate sedation was observed in one mule…’ is that referred to the epidural procedure? If this mule was nervous, would it not be excluded from the study given your assumption at LL.86-87 …. Was this mule not considered as intractable behaviour? ‘…. More time was required…’ remove IT; is there a mean reportable time that could be shown to better address this issue? was this the same mule that later on required further drugs also for the surgical procedure?

The sentence has been changed in order to make it more understandable: “The 1.3 mg/kg dose of IV xylazine was adequate in five out of the eight animals. The inadequate sedation was observed in one mule that received premedication IV before the catheter placement. As a result of the nervous behaviour, more time it was required for the catheter placement resulting in a reduction of the effect of the xylazine, as the half-life of this drug is 15 mins shorter (32 mins) in mules than in horses (47 mins) [30].This is in accordance with a previous study, demonstrating that mules may require approximately 50% more xylazine for sedation (and probably other alpha 2 agonists) than either donkeys or horses [1,28]. One mule jumped out the stocks during needle epidural placement, and this event may be a possible consequence during the procedure. One mule was a slightly obese and required a longer time to identify the first intercoccygeal vertebrae space”.

L 202. Please reformulate the sentence with ‘…possible effect…’, moreover your reference is not appropriate, as this is a review for equine standing laparoscopy in equine but there’s no real mention to possible side effects at the time of epidural placement, while the ‘jumping out’ is referred to skin clamp placement during draping. Did this mule receive the extra xylazine as well although its sedation was probably adequate according to scoring system of table 1?

Clarification added in the text.

  1. 204. ‘…is a painful procedure,…’ I would rather say that, ‘…although mini-invasive in nature, it is a painful procedure, especially when...’ '..is grasped and ligated'....which technique did the authors used for ovariectomy?, ‘…or simply topical anaesthesia’ [12,28-30]

Added in the text: “Although standing laparoscopic ovariectomy is a mini-invasive in nature, itis a painful procedure, especially when the pneumoperitoneum was induced and when the ovarian pedicle is grasped and ligated. For  these reasons, a slowly carbon dioxide insufflation and an injection of local anaesthetics in the ligament, mesovarium and mesosalpinx or topical anaesthesia are recommended [4,12,32-34]. In this study, epidural anaesthesia allowed ovarian manipulation in 6 out of 8 mules. In one mule a supplemental IV dose of xylazine 0.65 mg/kg was sufficient to complete the surgery, while in one mule it was no sufficient and infiltration of the ovarian pedicle with 20 ml of lidocaine 2% was needed to complete the surgery”.

  1. 207: is ‘Very few papers…’ referred to n.31 and n.32?? please include them after ‘…in mules’.

Included as suggested

  1. 218 I would enter the paragraph which start with ‘In this study,…’ and say ‘In the present study,…’

Modified as suggested

  1. 220-223. ‘…without infiltrating local…’ rather than instilling, however, at lines 174-177 you described that only 1 mule required 20 ml lidocaine into the ovarian pedicle, here they are 2, please amend where required. ‘…while in two the manipulation….’ are those other 2 mules? or it is the same concept?

Concept clarified as follows: “In this study, epidural anaesthesia allowed ovarian manipulation rather than instilling local anaesthetic into the ovarian pedicle in 6 out of 8 mules, while in two the manipulation of one ovary was not completely tolerated. Therefore, one mule received a supplemental IV dose of xylazine 0.65 mg/kg to complete the surgery, while the other one the xylazine bolus was no sufficient and the ovarian pedicle was infiltrated with 20 ml of lidocaine 2% to complete the surgery”.

221-223. This concept is very interesting but not very clear, you mention that the body length was greater (weren't mules similar in weight as you stated in your results?) was the body weight similar instead? The body weight was similar but the body length of two mules was greater.

So that operators should pay attention more to that physical parameter rather than to the effective weight in mules, which is the main parameter we normally consider when preparing the dose and volume for epidural combinations, ‘..epidural analgesic drugs combination…’

However, I wonder how L3 could be reached with your caudal epidural approach with only 10 ml volume. Please verify your assumption, as normally maximal dermatomal analgesic spread ranges from C1-S3 spinal cord segments with caudal epidural injection. It was a typo (L3 instead S3.)

  1. 226: ‘…vertebral column, as IT has been reported… [34-36]’.

Modified as suggested.

  1. 233: ‘In this study, two animals also showed lack of analgesia following …’ why also?

“also” has been deleted.

Conclusion

This part could be improved, moreover, don’t forget to mention that your epidural was combined with a continuous rate infusion of xylazine (which has not been so largely described in mules)

Please amend abbreviation for Coccygeal vertebrae (Co)

Conclusion has been modified as suggested.

References

In general, there are some typos in your references.

n.1 Matthews, then a comma

n.5 check the name of the second author

n.6 check the name of the first author

n.7 check adequacy for journal

n.9-10 Comma after the surname

n.11 check names

n.14 check names

n.17 check format for journal writing

  1. 19 check name second author

n.20 check format

n.22 comma after DeRossi and full stop after Sci

n.23 space after 2019

n.27 Latzel,

n.28 check format

n.30 check (lidocaine in)

n.32 mares; check reference details for Volume and format

n.34 verify adequacy from guidelines

n.35 check writing format

n.36 check writing format

References have been verified as suggested.

Reviewer 3 Report

Thank you for your submitted article to this journal.

The information about laparoscopic surgery to the mule mares was very interested to our veterinary anesthesiologists and veterinary clinicians.

I judged that your article content and research style are also very good.However, I would like to point out some questions and improvements.

Author Response

we thank very much for your constructive comments. As you suggested, we tried to improve the quality of the manuscript accordingly. As requested, all changes made to the manuscript, in comparison to the previous version, have been reported in the text using “track changes” in Microsoft Word.

Below, the answers of the authors to the comments.

Analgesic and sedative effects of epidural lidocaine-xylazine in elective bilateral laparoscopic ovariectomy in standing mule mares

L52, and L55 (reference no.8,9,10)
I couldn't get any detailed information about “reduced morbidity and mortality”, and I have some doubts about just extrapolating as like horses.

We agree with you, and we have changed the sentence to the following statement: One advantage of standing sedation over general anesthesia is the decreased risk of mortality [8]. Benefits of standing laparoscopy, compared to laparotomy performed under general anaesthesia, include lower costs, direct visualization of abdominal viscera, secure hemostasis, tension-free manipulation of the ovaries and smaller flank incisions. These advantages reduce complications and hospitalization-time associated with ovariectomy [9,10].

L64-65
The expression “C1-C2” confused “C”ervical. So I thought it would be better to change it as “Ca 1- Ca2”.

OK

L89 ; fasting time
Is it popular too long fasting time for mule ?

Sorry, It was a typo, we’ve changed it to 16 hours

L93; stock and the tails were wrapped
Please add information such as the product name and material used for the wrap.

Information added.

Table 1 and 2;
Please specify who did these scoring. Ensuring the objectivity of these scores is a very important point in this study.

Specified as requested.

Score; analgesia
I think it's better to change to the notation “ Analgesic score “(L127, in the Table 2)

Modified as requested.

L128; inadequate
Please describe in detail what kind of state was judged as "inadequate"

Detailed description has been made in the text.

L129; responded
Why didn’t you used the heart rate, respiratory rate and invasive blood pressure fluctuations as response indicators?
I would like to consider adding an evaluation of these changes in biometric information.

In order to evaluate the depth of sedation / analgesia during this type of surgical procedure, in this species, it was more immediate and effective to use the response of the animal to the surgical stimulus using the scores described in Table 2. Despite having taken these into consideration, heart rate, respiratory rate and arterial blood pressures, they did not show significant fluctuations directly related to the surgical stimulus. Therefore, they were considered less suitable for this purpose.

In the table2; Sedation depth
You mentioned at L131 “depth of sedation” and Fig.1 “SEDATION”.
Notation should be unified. And please clarify the difference between Table-1 depth of sedation and Table-2 Sedation depth and Fig.1 SEDATION.

Thank you for the comment. Notation was unified to “Depth of Sedation” both in Table 2 and in Figure 1. The sedation scale described in Table 1 was used to assess the depth of sedation before epidural administration (and therefore before surgery). The sedation scale described in Table 2 was used to assess the depth of sedation during surgery (and therefore after epidural anaesthesia) by evaluating the response of the animal to the surgical stimulus. The scoring system described in Table 2, in addition to the depth of sedation, also included the assessment of analgesia and ataxia. The scale described in the M&M section with Table 2 was used to obtain the results presented in Figure 1. We hope that it is now better described and discussed in M&M and Discussion sections.

In the Table-3
What did you indicate a, b, c, etc. in teach culumn? I would like an explanation in the annotation.

The caption of Table 3 has been modified to make the meaning of the letters and numbers in superscript more understandable. Letters and asterisks indicate differences at post-hoc multiple comparison tests. The pair of equal letters (e.g. a-a, b-b, and so on) indicate which are the pairs of time-points showing significant differences; the asterisks indicate the entity of each difference as described in the annotations.

L170〜
Do the sequelae observed here only occur with a laparoscopic surgery? If so, I would like you to consider preventive measures, etc., rather than concluding that you have healed naturally.

The treatment used has been added.

L177〜
Please discuss in a little more detail why it seems that no statistical difference was found.

To assess whether the frequency of finding insufficient sedation/analgesia during the study (i.e. the number of mules requiring supplemental xylazine and/or infiltration of the ovarian pedicle with lidocaine) was statistically relevant, a frequency analysis was performed considering all sequential time points. Epidural anaesthesia allowed ovarian manipulation in six mules (out of eight) while in two the manipulation of the ovary was not completely tolerated. In these two mules, an additional administration of xylazine was sufficient in one mule, while in the other the xylazine was not sufficient but it was necessary to infiltrate the ovarian pedicle with lidocaine. The statistical results of the Chi-square (χ2) test for trend showed no significant differences considering the frequency trend of mules requiring supplemental xylazine or lidocaine infiltration during surgery. A more detailed discussion of this aspect was added in the Discussion section.

L202; One mule jumped out the stocks....
If even one animal experiences such a situation, I think it is a big problem in the anesthesia management of livestock. What are your thoughts on the comprehensive evaluation of the safety of the method used this time? Please add to the discussion and conclusions.

We added in the discussion: One mule tried to jump out the stock during needle epidural placement, and this event may be a possible consequence during the procedure. This event could be a big problem, therefore during the epidural anaesthesia the operator should pay attention to confine the animals in a dedicated room.

Round 2

Reviewer 1 Report

As I mentioned before, the paper is concise and objective, and even so I observe that it has improved a lot in its written form.
I understand that many works or projects involving animals have the difficulty of working with a significant enough number for statistical analysis, or even completing all groups
After reflecting on its significance, and especially that the procedure was successful, I am in favor of publishing it even without the control group.

Att.